# Centrifugal Atomization of Glass-Forming Alloy Al_86_Ni_8_Y_4.5_La_1.5_

**DOI:** 10.3390/ma15228159

**Published:** 2022-11-17

**Authors:** Jordi Pijuan, Sasha Alejandra Cegarra, Sergi Dosta, Vicente Albaladejo-Fuentes, María Dolores Riera

**Affiliations:** 1Eurecat, Centre Tecnològic de Catalunya, Unit of Metallic and Ceramic Materials, Plaça de la Ciència 2, 08243 Manresa, Spain; 2Departament de Ciència dels Materials i Química Física, Universitat de Barcelona, Martí i Franqués 1, 08028 Barcelona, Spain; 3Thermal Spray Centre (CPT), Departament de Ciència dels Materials i Química Física, Universitat de Barcelona, Martí i Franqués 1, 08028 Barcelona, Spain; 4Department of Mining, Industrial and ICT Engineering, Universitat Politècnica de Catalunya, Av. De les Bases de Manresa, 61-73, 08242 Manresa, Spain

**Keywords:** metallic glasses, centrifugal atomization, amorphous fraction, cooling rate

## Abstract

Centrifugal atomization is a rapid solidification technique for producing metal powders. However, its wide application has been limited to the production of common metal powders and their corresponding alloys. Therefore, there is a lack of research on the production of novel materials such as metallic glasses using this technology. In this paper, aluminum-based glassy powders (Al_86_Ni_8_Y_4.5_La_1.5_) were produced by centrifugal atomization. The effects of disk speed, atomization gas, and particle size on the cooling rate and the final microstructure of the resulting powder were investigated. The powders were characterized using SEM and XRD, and the amorphous fractions of the atomized powder samples were quantified through DSC analysis. A theoretical model was developed to evaluate the thermal evolution of the atomized droplets and to calculate their cooling rate. The average cooling rate experienced by the centrifugally atomized powder was calculated to be approximately 7 × 10^5^ Ks^−1^ for particle sizes of 32.5 μm atomized at 40,000 rpm in a helium atmosphere. Amorphous fractions from 60% to 70% were obtained in particles with sizes of up to 125 μm in the most favorable atomization conditions.

## 1. Introduction

Metallic glasses have attracted the interest of the scientific community for their possible applications due to their unique properties over their crystalline counterparts. Within amorphous metal alloys, Al-based metallic glasses have gained attention since their discovery in 1988 [1,2] because of their high specific strength [3,4] and high resistance to corrosion [5] compared to general-use aluminum alloys, which makes these materials an important target for their potential development in engineering applications.

Al-based metallic glasses have been proven to be very difficult to obtain due to their low glass-forming ability (GFA) compared to other metallic glasses [6,7]. The vast majority of the studied alloys are within the AL-RE-TM (aluminum, rare earths, and transition metals) system because they exhibit a higher GFA [8]. Yang et al. [4] produced glassy rods of 1 mm in diameter using copper mold casting, which is the largest product reported for Al-based metallic glasses with a composition of Al_86_Ni_6_Y_4.5_Co_2_La_1.5_.

Low GFA suggests a higher critical cooling rate, therefore limiting the sample size that can be achieved in a fully amorphous structure. The cooling rate necessary to obtain amorphous Al-based alloys is in the order of 10^3^–10^6^ K s^−1^ [9]. Thus, the most viable way to manufacture these metallic glasses are through rapid cooling processes, with the most common including melt spinning [10] and gas atomization [11,12,13]. Although melt spinning is popular for scientific research, the maximum achievable size for specimens is restricted [14]. On the other hand, for gas atomization, it has been found that the largest completely amorphous particle size obtained for these alloys are powders of less than 25 μm [15,16].

Although the resulting product of atomization methods is a powder, these techniques have maintained scientific interest due to the possibility of producing amorphous or partially amorphous powder that can be further processed using various technologies. Al-based metallic glass powders have been successfully deposited by cold gas spraying, demonstrating that is possible to maintain amorphous structure and to create functional coatings with high hardness and corrosion resistance [5,17,18]. Additionally, selective laser melting [19,20] and spark plasma sintering [21] had been demonstrated to be good techniques to maintain a partially amorphous structure of consolidated bulk specimens, but studies on Al-based amorphous alloys in these fields are still incipient.

Today, the production of Al-based metallic glasses is still challenging. An alternative technique to gas atomization for generating these materials is centrifugal atomization. During this process, molten metal is dropped onto a high-speed rotating disk. The liquid spreads out onto the disk by the centrifugal force and finally breaks into small droplets that solidify during their flight in the presence of an inert gas atmosphere. Cooling rates between 10^3^ and 10^5^ K s^−1^ can be reached using this method [22], thus providing an appropriate range of cooling rates for the generation of Al-based metallic glasses. However, a review of the literature reveals that although there are several works on the centrifugal atomization process [23,24,25], studies on the production of metallic glasses using this technique are not found in the literature.

In addition, centrifugal atomization has significant advantages compared to traditional gas atomization technology. The particle size distribution of the batch produced is narrower, resulting in a higher yield process. The powder that is produced has a highly spherical form, with better roundness and a smooth surface. Unlike gas atomization, satellites, which are adhesions of small particles on larger ones, are minimized. All of these powder characteristics significantly improve powder flowability, a key factor in additive manufacturing or thermal spraying technologies. Additionally, inert gas consumption and the energy consumption associated with it is significantly reduced, making centrifugal atomization highly beneficial in terms of sustainability and cost reduction.

In the present work, an Al_86_Ni_8_Y_4.5_La_1.5_ alloy was centrifugally atomized, and its microstructure and the amorphous fraction that were obtained were investigated as a function of gas composition and rotational disk speed. Al_86_Ni_8_Y_4.5_La_1.5_ was chosen because it showed a strong capacity in the formation of metallic glasses, as derived from an extensive study of aluminum-based bulk metallic glasses [26]. However, the centrifugal atomization of the powder resulting from this composition has never been reported on before. This research is aimed at characterizing the glass formation ability of the atomized Al_86_Ni_8_Y_4.5_La_1.5_ and at expanding our understanding of metallic glass preparation via the centrifugal atomization technique.

## 2. Materials and Methods

### 2.1. Powder Synthesis

A schematic diagram of the centrifugal atomization unit is shown in Figure 1. This equipment consists of three main parts: an atomization chamber to provide a controlled atmosphere; an induction melting system with a crucible that allows the metal to be melted in an inert gas atmosphere; and a spinning disk, which disintegrates the melt into fine particles.

The raw materials used for alloy formation were commercial-purity Al-Y master alloy (90%Al-10%Y mass%), Y-La lump (12%Al-88%Y mass%), La-Ni eutectic pieces (12%Ni-88%La mass %), and commercially pure nickel 201 alloy (>99 Ni mass%). They were mixed in a molar composition of Al_86_Ni_8_Y_4.5_La_1.5_ and then induction-melted at 1605 K in an alumina crucible equipped with a stopper rod that allowed the melt to flow through the crucible orifice. The temperature was measured by means of a thermocouple located inside the crucible and was kept constant for 15 min to ensure homogenization and proper dissolution of all of the raw materials. Prior to melting and atomization, a vacuum pump was run to depressurize the chamber to 10^−3^ Pa and was subsequently backfilled with inert gas. During the atomization process, the stopper rod was pulled up, and the melt was gravity poured through the 3 mm orifice onto a flat 40 mm diameter 316 L stainless steel disk rotating at a high speed. This disk had a 150 μm ZrO_2_ coating on the upper surface in contact with the melt. A water-cooling system was used to guarantee the integrity of the disk due to the high temperatures of atomization.

The main processing parameters for this study were the disk speed and the inert gas atmosphere. A set of test runs were carried out, melting a total of 350 g of the starting materials for each experiment. Process conditions consisted of analyzing two different disk speeds: 30,000 rpm and 40,000 rpm; and three combinations of different gas compositions (in volume %): 100% Ar (Ar), 50%He-50%Ar (He-Ar), and 100% He (He).

### 2.2. Powder Characterization

The powder obtained from each atomization run was collected and sieved according to ASTM-B215 for 15 min with a Filtra FTL-0150 electromagnetic digital sieving machine with 20, 45, 75, 106, 125, 150, and 180 μm sieves according to the corresponding particle fraction sizes.

The particles were then embedded in cold-mounting resin, ground, and polished with diamond suspension solution and finished with a colloidal dispersion of silica (SiO_2_) using standard metallographic methods for uncompacted metal powders [27]. The surface morphology and microstructure of the powders were investigated using an Ultra Plus field-emission scanning electron microscope (SEM). The chemical compositions of the atomized powders were checked by inductively coupled plasma optical emission spectroscopy (ICP-OES) using Thermo Scientific iCAP PRO. X-ray diffraction analysis of the sieved Al alloy powder was performed in order to verify the production of amorphous material, even for high-size powder fractions. XRD patterns were recorded using PANalytical X’Pert PRO MPD equipment with Cu Kα radiation (λ = 1.5418 Å) from 5 to 100° 2θ with a 0.017° step, measuring 100 s per step. The differential scanning calorimetry (DSC) technique was selected to determine the volume fraction of the amorphous (*V_f_*) phase present in the different powder fractions produced by centrifugal atomization. For this analysis, a DSC1 Mettler Toledo was used, and the measurements were conducted under a nitrogen atmosphere using aluminum crucibles. For the test, 20 mg of each powder was heated from 300 K to 800 K at 10 K/min rate.

## 3. Results

A summary of the test runs performed, their atomizing conditions, and the particle size distribution of the obtained powders is presented in Table 1. The particle size distributions of the atomized powders at 40,000 rpm are finer than those atomized at 30,000 rpm. This result is coherent with the known state-of-the-art of centrifugal atomization [28], as the higher the disk velocity, the lower the particle size that can be achieved. Regarding the gas atmosphere used in atomization, no clear tendency was found in the particle size distribution results, allowing us to conclude that the nature of the gas does not affect the particle size distribution.

Chemical analysis by inductively coupled plasma (ICP) of the final powder indicates that the alloy composition is close to the desired nominal composition. The final composition of the powder was Ni:8.1, Y: 4.6, La: 1.6, Al: bal. (in atomic %).

### 3.1. SEM Analysis

SEM images of the powders in the particle size range of 75–45 µm atomized under different conditions are shown in Figure 2. Most of the powder particles were spherical in shape. The surfaces of particles atomized in a He atmosphere were very smooth, and direct observations did not show a crystalline structure, whereas particles atomized in He-Ar—and to a larger extent, in Ar atmospheres—presented rough surfaces that are related to the formation of crystalline phases.

Micrographs of centrifugally atomized particle sizes synthesized in a He atmosphere are shown in Figure 3. Figure 3a shows the cross-section micrograph of the 45–20 μm particle size range, where the majority of the particles have a featureless structure. In some particles, small Al_2_Y crystals (less than 10 μm in length) surrounded by the featureless structure appear [15]. It is important to notice that for the smaller particle sizes, it cannot be determined if the particles are completely featureless. SEM observations only show the cross section of the particles, meaning that they may have a certain degree of crystallization that remains unknown.

A similar pattern is detected in the micrographs of the 75–45 μm particle range. In Figure 3b, most of the particles also appear to be featureless, but some of them are clearly crystallized, with high-length crystals and a heterogeneous crystallographic structure that is typical of the crystallization of Al-Ni-Y alloys [12,16]. High randomness is noticed in the microstructure between particles for the same particle size range and atomization conditions, in which some seem totally amorphous, and others are almost totally crystalline. Despite this disparity, a clear pattern appears in the proportion of featureless and crystallized particles depending on particle size and atomization gas. In small particle size ranges and in particles atomized in He, the majority of the particles present a featureless structure, while in large particle size ranges atomized in He-Ar and Ar, almost the totality of the particles are fully crystallized.

### 3.2. XRD Analysis

XRD patterns of centrifugally atomized powder samples at different gas compositions and disk speeds are shown in Figure 4. Phase identification in Figure 4 is based on results from [12]. From these diffraction patterns, it can be verified that the crystalline fraction increases with the particle size, even for different gas compositions, as previously observed in the micrographs.

When comparing the diffraction patterns of particles atomized in He and Ar, it is clear that particles atomized in an Ar atmosphere have more crystallinity than particles atomized in a He atmosphere. A broad amorphous halo was observed in the particles atomized in a He atmosphere at a disk speed of 40,000 rpm, indicating the formation of an amorphous structure up to a particle class size of 125–106 µm. However, even for small particle sizes (45–20 µm), small diffraction peaks close to the background signal indicate that these particles are not fully amorphous. Peaks appear more clearly as the particle size increases, indicating an increase in the crystalline fraction in larger particles. These peaks correspond to the α-Al phase [29]. In comparison, the XRD patterns of particles atomized in an Ar atmosphere at a disk speed of 30,000 rpm show clear diffraction peaks, even in particle sizes of 45–20 µm. For particles > 75 µm atomized in an Ar atmosphere, the DRX results correspond to an almost entirely crystalline pattern, as no halo is identified in these cases.

### 3.3. DSC Analysis

Figure 5 shows a DSC measurement from room temperature to 800 K (10 K min^−1^) for a 45–75 µm particle size range sample atomized in a He atmosphere at a disk speed of 40,000 rpm. DSC curves for the different gas compositions and the same particle range are shown in the inset in Figure 5. The DSC curve of the ribbon sample is adapted from [26]. For all of the atomization conditions, the DSC curves exhibited a three-stage crystallization process characterized by three peak crystallization temperatures, as expected for this alloy. No detailed analysis of the crystallization of the Al_86_Ni_8_Y_4.5_La_1.5_ glass-forming alloy is found in the literature. However, other similar compositions are studied in detail. The first peak corresponds to nano-Al precipitation, the second peak corresponds to the growth of alpha-phase nanocrystals and the formation of Al_3_Ni and Al-Ni-Y phases, and the third peak corresponds to the growth of existing crystals and to the formation of Al-La phases [4,30]. No sign of glass transition temperature (*T_g_*) was detected from the DSC curve. For most of the Al-based metallic glasses, the supercooled liquid region is small; therefore, the glass transition signal and the onset of the crystallization signal tend to overlap [5,31].

### 3.4. Determination of Amorphous Fraction

The volume fraction of the amorphous phase (*V_f_*) in each atomized particle size range was evaluated by normalizing the enthalpy released during the crystallization of the powder with the enthalpy of crystallization of a fully amorphous melt-spun ribbon sample using the following equation:(1)$$V_{f}=\left(\Delta H_{T,Am}-\Delta H_{T,\;PAm}\right)/\Delta H_{T,\;Am}$$
where Δ*H_T,Am_* is the total enthalpy released upon the crystallization of a fully amorphous sample, and Δ*H_T,PAm_* is the total enthalpy released upon the crystallization of a partially amorphous sample. For this work, the enthalpy of the crystallization of the fully amorphous melt-spun ribbon sample was taken from Yang et al. and used as a reference [26].

The heat released during the three exothermic reactions was compared with the heat released from the three exothermic reactions of the fully amorphous Al_86_Ni_8_Y_4.5_La_1.5_ melt-spun ribbon of the identical crystallization sequence. Figure 6 gathers the total amorphous volume fraction present in the atomized particles as a function of the corresponding size range for the different conditions of atomization.

Figure 6a shows that for a given particle size range, centrifugally atomized powders prepared in a He atmosphere have a much higher amorphous fraction than powders atomized in an Ar and He-Ar atmosphere, whereas Figure 6b shows that changes in the disk speed do not seem to have a significant effect on the achieved amorphous fraction compared to the gas composition. From Figure 6, it can be seen that centrifugally atomized powders prepared in a He atmosphere, either at 30,000 rpm or 40,000 rpm, have a high amorphous fraction of 60% to 70% in particle size ranges of up to 125–106 µm. To the authors’ knowledge, there are few data on amorphous volume fractions in atomized alloys in the existing literature related to the Al-TM-RE system [5,12,13]. The literature suggests that a high percentage of amorphous fractions have only been found for particle sizes <40 µm for particles atomized via gas atomization, even for those atomized in a He atmosphere.

### 3.5. Thermal Evolution of Atomized Droplets

A mathematical heat transfer analysis was used to quantify the heat transport between the surrounding gas and the centrifugally atomized droplets. Particle trajectories along the atomization chamber were computed according to Yule [32] by considering drag force and gravity force and by assuming an initial velocity equal to the tangential velocity of the disk [33]. A homogeneous temperature distribution inside the droplet is assumed to be due to rapid heat conduction within the particle compared with convection heat transfer from the surface of the droplet and the surrounding gas [34]. Radiation is considered to be negligible due to the low contribution on the cooling rate compared to convection, which is around an order of magnitude lower [11,32]. To calculate the effective heat transfer coefficient *h*, the Nusselt number *Nu* from the semi-empirical equation based on the Whitaker correlation [34] was used [34], which considers the temperature-dependent thermophysical properties of the gas in an environment with a high temperature gradient [35]:(2)$$Nu=h\;d/k_{g}=2+\left(0.4Re^{1/2}+0.06Re^{2/3}\right)Pr^{1/4}{\left(\mu_{g}/\mu_{s}\right)}^{1/4}$$
where *d* is the droplet diameter, *k_g_* and *µ_g_* are the thermal conductivity and the viscosity of the gas evaluated at ambient temperature, respectively, and *µ_s_* is the gas viscosity evaluated at the temperature of the surface of the droplet. *Nu* includes the Reynolds number *Re* and the Prandtl number *Pr,* which are defined as:(3)$$R_{e}=\rho_{g}\;v\;d/\mu_{g}$$
and
(4)$$P_{r}=C_{pg}\;\mu_{g}/k_{g}$$
where *C_pg_* and *ρ_g_* are the specific heat capacity and density of the gas evaluated at the ambient temperature inside the atomizer, respectively, and *v* is the droplet velocity.

For simplicity, the thermophysical properties for pure Al liquid were used for the Al_86_Ni_8_Y_4.5_La_1.5_ alloy. The glass transition temperature *T_g_* of the Al_86_Ni_8_Y_4.5_La_1.5_ alloy was taken as 507 K [26]. Since the experiment involved metallic glass powder, this calculation was performed by assuming that no phase change takes place during the cooling of the molten metal from the atomization temperature to the glass transition temperature *T_g_* [11,36]. Thus, for the cooling rate calculation, it was assumed that undercooling takes place below the glass transition temperature.

Figure 7 shows the cooling evolution of the atomized droplets from the atomization temperature to the glass transition temperature in the different study cases. Droplets from the lower diameter are cooled more rapidly, and He provides the best cooling rate. Comparing the cases of equal atmosphere and different disk velocities, only a slight improvement in the cooling rate is observed at higher velocities. In the case of He atomization, all of the droplets reach the glass transition temperature in 20 ms, while in a He-Ar atmosphere, only particles of around 100 μm in size achieve this cooling time, and only finer particles below 50 μm reach this condition in an Ar atmosphere. The largest time interval is 150 ms for droplets of 165 μm in size atomized in Ar.

## 4. Discussion

### 4.1. Cooling Rate Calculation

The cooling rate of the atomized droplets, assuming no phase change appears before reaching the glass transition temperature, can be evaluated as [32]:(5)$$\dot{T}=\frac{6h(T_{d}-T_{\infty })}{C_{pl}d}$$
where *T_d_* is the droplet temperature, *T_∞_* is the temperature of the gas atmosphere, and *C_pl_* is the specific heat capacity of the droplet. In Figure 8, the values of the cooling rate over time for different particle sizes and atomization conditions are represented. These plots represent the cooling rate experienced by the particles from the initial melt temperature to the glass transition temperature *T_g_*. The case studies for 30,000 rpm are not shown due to similarity with the 40,000 rpm cases. As mentioned before, He provides larger cooling rates, and Ar provides the lowest cooling rates, and finer droplets are also cooled more rapidly.

A droplet of 33 μm in a He atmosphere experiences a cooling rate between 2 × 10^6^ Ks^−1^ and 1.5 × 10^5^ Ks^−1^ from the melt temperature to the glass transition temperature. In the case of an Ar atmosphere, these values are reduced by approximately an order of magnitude. Comparing the different atmosphere gas cases, droplets of 137.5 μm have a cooling rate approximately an order for magnitude lower than 33 μm droplets. Even finer particles of 33 μm atomized in an Ar atmosphere have a higher cooling rate than large particles of 137.5 μm atomized in a He atmosphere do.

### 4.2. Comparison between Cooling Rate and Amorphous Fraction

In order to obtain comparable and interpretable results between the cooling rate and the amorphous fraction, an average cooling rate is preferred for the entire cooling process *CR* instead of $\dot{T}$, and this is evaluated as [36]:(6)$$CR=\frac{T_{L}-T_{g}}{t_{L}-t_{g}}$$
where *t_L_ − t_g_* is the time between the liquidus temperature *T_L_* and the glass transition temperature *T_g_*.

Figure 9 illustrates the average cooling rate *CR* as a function of particle size with the corresponding amorphous fraction obtained from the DSC data for different gas compositions. The amorphous fraction decreases rapidly for cooling rates below 6 × 10^4^ Ks^−1^ for all the atomization conditions.

The maximum cooling rate is achieved in a He atmosphere, where particles of 32.5 µm experience an average cooling rate of 7 × 10^5^ K s^−1^. In an Ar atmosphere, the cooling rate decreases an order of magnitude compared to the cooling rate achieved in a He atmosphere for the same particle sizes. Particles of 32.5 µm have an average cooling rate of an order magnitude higher than particles of 165 µm atomized in the same atmosphere. Hence, it is concluded that He has a stronger effect than Ar, resulting in a higher cooling rate, which is in good agreement with the literature [36,37]. This is mainly due to the thermal conductivity of He being greater than that of Ar (1.52 × 10^−1^ Wm^−1^ k^−1^ value for He and 1.77 × 10^−2^ Wm^−1^ k^−1^ for Ar, both at 300 K, which is the initial temperature of the gas in the atomization chamber). The particle diameter also has a strong influence on the cooling rate and is as important as the gas atmosphere.

In a case where the same cooling rate is taken for any of the atmospheres, a similar amorphous fraction is obtained independently of the particle size diameter. For example, a similar cooling rate and a similar amorphous fraction are obtained for 32.5 µm particles atomized in Ar and 120 µm particles atomized in He. The same pattern is found when comparing particles atomized in a He atmosphere that are 150 μm in size and particles atomized in Ar that are 60 μm in size, whereas in this case, both appear to be almost fully crystalline.

### 4.3. Effect of Gas Composition in Amorphous Fraction

The cooling rates necessary to achieve completely amorphous microstructures in Al-based metallic glasses have been calculated theoretically for different techniques such as gas atomization and melt spinning [11,38]. In both cases, the cooling rate was estimated to be 10^6^ Ks^−1^. Figure 9 shows that in a He atmosphere, although particle sizes of 45–20 μm achieved a cooling rate of near 10^6^ Ks^−1^, the amorphous fraction obtained in this work is below 70%. However, in a He atmosphere, the amorphous fraction value is kept relatively stable, with a slight decrease as the droplet diameter increases in particles of up to 125 μm in size, after which the amorphous fraction decreases rapidly. In a He-Ar atmosphere, this rapid decrease appears at particle sizes of 90 μm and at around 40 μm in an Ar atmosphere. Comparing these results with time intervals in Figure 7, a significant amorphous fraction is obtained in all cases if the particles are cooled in at least 15 ms, which represents an average cooling rate of 7.3 × 10^4^ Ks^−1^. This cooling rate value is larger than those experimentally determined in a laboratory with melt spinning, which are between 3 × 10^3^ and 10^4^ Ks^−1^ [9].

The obtention of particles that are not fully amorphous but that achieve the required cooling rates may be due to impurities or oxide particles from the raw material that act as nucleation sites that favor the crystallization of the final powder [39]. Consequently, it is difficult to obtain fully amorphous powders since these impurities may cause crystalline phases. Commercial starting materials and commercial alumina crucibles were used in atomization experiments, and these may be the result of a higher presence of impurities in the atomized melt. The presence of impurities could also justify the wide variability in the particle microstructure shown in Figure 3b depending on whether the presence of an impurity acts as nucleation point in the atomized droplet.

Although a fully amorphous fraction has not been achieved in any particle size range, a significant amorphous fraction is obtained in particle sizes of up to 125 μm atomized in a He atmosphere. This demonstrates that centrifugal atomization is able to achieve high cooling rates, even in large particles.

Another aspect that can contribute to crystallization is melt superheat. Some investigations have concluded that a lower melt superheat temperature contributes to the cooling path not reaching the crystallization zone in the time–temperature–transformation diagram [40]. A high melt superheat of 1605 K was used in the atomization runs (400 K above liquidus temperature) to guarantee the complete melting of the starting materials and to avoid possible melt solidification in the atomization disk. In this study, the cooling effect of the atomization disk in the melt is an aspect that has been assumed to have no significant influence, but detailed analysis should be required to estimate the initial temperature of the droplet when it is expelled from the atomization disk.

### 4.4. Effect of Disk Speed in Amorphous Fraction

Particles atomized at different disk speeds do not show a significant change in the microstructure. Theoretical results show a minor decrease in the cooling rate at lower disk speed, and there is also a slight reduction in the amorphous fraction for particles atomized at 30,000 rpm compared to particles atomized at 40,000 rpm. The experimental and theoretical results show that disk velocity does not contribute to an increase in the cooling rate of the particles. An increase of 33% in the disk velocity, and therefore in the droplet initial velocity, while contributing to the increase in the heat transfer coefficient between the droplet and the surrounding gas, has no significant effect on the cooling rate compared to other variables such as droplet diameter and the gas atmosphere. Nevertheless, disk velocity will have a strong influence on obtaining finer droplets and will contribute to the overall yield to obtain powders with a higher amorphous fraction.

## 5. Conclusions

Powder samples of Al_86_Ni_8_Y_4.5_La_1.5_ glass-forming alloy were atomized by centrifugal atomization technology. The influence of processing parameters such as disk speed and gas composition was studied. Most of the powders were spherical in shape, while the surface morphology was different for the finer and the coarser powders. Powders of particle sizes below 45 μm mainly had a smooth morphology where no optically detectable microstructure was identified. However, as the particle size increased, partially crystalline and fully crystalline microstructures were observed.

The volume fraction of the amorphous phase for various particle size ranges were determined from DSC analysis. Centrifugally atomized particles synthesized in a He atmosphere resulted in an amorphous fraction of around 70% for particle sizes <45 μm and between 50% and 65% for particle sizes <125 μm. However, particles atomized in an Ar atmosphere resulted in an amorphous fraction of 55% for particle sizes <45 μm, and this value decreased sharply as the particle size increased. Limitations in achieving a fully amorphous fraction might be due to the presence of nucleants in the melt during the atomization process.

The cooling rate experienced by centrifugally atomized powders was determined theoretically. As expected, the cooling rate increases as the particle size decreases. A cooling rate of 7 × 10^5^ Ks^−1^ for particles of 32.5 μm atomized in a He atmosphere was obtained, and this value was reduced by one order of magnitude for particles atomized in an Ar atmosphere. As for the disk speed, no significant increase in the amorphous fraction was observed as the disk speed increased from 30,000 rpm to 40,000 rpm.

The results show that the centrifugal atomization technique is able to achieve cooling rates that are high enough to obtain an amorphous fraction above 50% in particle sizes of up to 125 μm using a He atmosphere. Although more research is necessary to achieve a higher amorphous fraction, it is demonstrated that centrifugal atomization is a viable technique for obtaining amorphous powders of larger particle sizes in alloys with a low glass-forming ability, such as Al-based metallic glasses.

## Figures and Tables

**Figure 1 materials-15-08159-f001:**
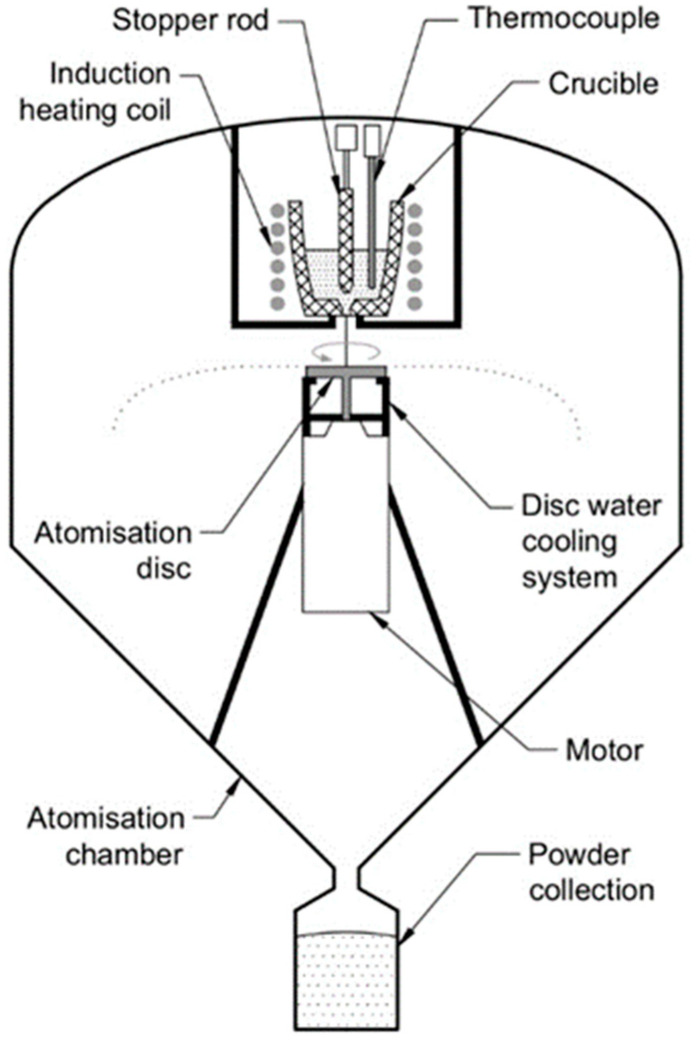
Schematic diagram of the centrifugal atomization equipment.

**Figure 2 materials-15-08159-f002:**
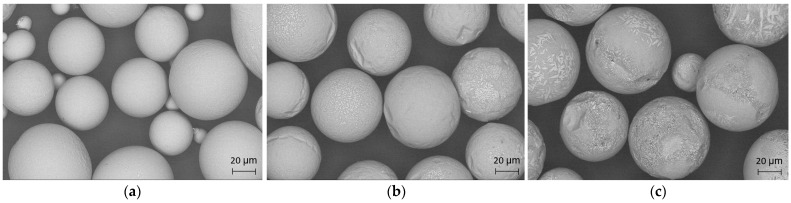
Surface morphology of the resulting powder in the size range of 75–45 µm atomized at the same speed of 40,000 rpm using different gas compositions: (**a**) He; (**b**) He-Ar; (**c**) Ar.

**Figure 3 materials-15-08159-f003:**
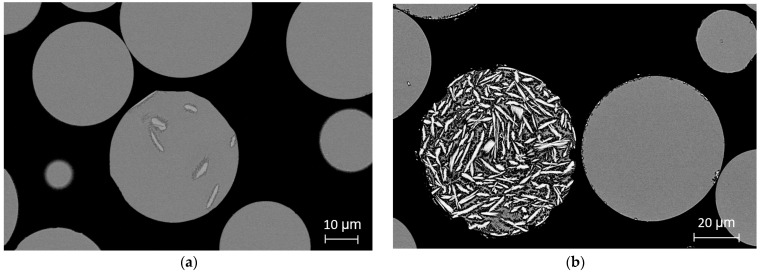
Cross-section SEM micrographs of particles centrifugally atomized in a He atmosphere at 40,000 rpm: (**a**) 45–20 μm; (**b**) 45–75 μm.

**Figure 4 materials-15-08159-f004:**
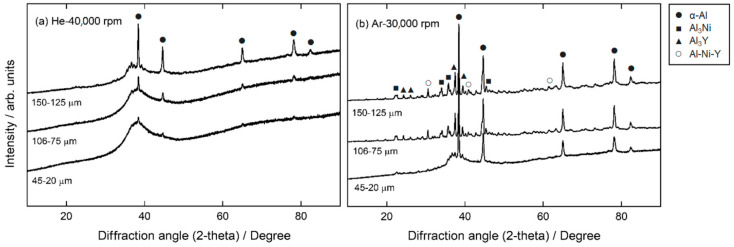
XRD pattern for centrifugally atomized particles: (**a**) particles atomized in a He atmosphere at 40,000 rpm, and (**b**) particles atomized in an Ar atmosphere at 30,000 rpm. The intensity of the crystalline peak increases with increasing particle sizes, indicating a higher crystalline fraction.

**Figure 5 materials-15-08159-f005:**
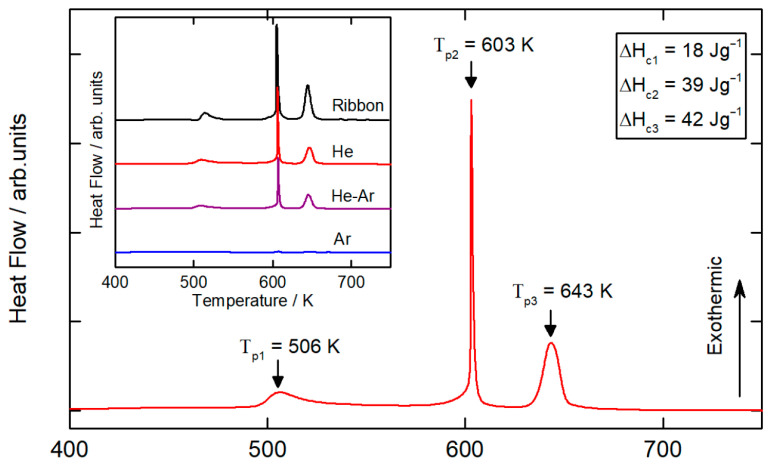
DSC measurements for 75–45 μm particles atomized in a He atmosphere at 40,000 rpm. ΔH_c1_, ΔH_c2_, and ΔH_c3_ are the integrated enthalpies of crystallization for the first, second, and third peaks, respectively. Inset in the figure shows the DSC curves of the respective samples atomized in different gas compositions, and DSC curve of ribbon sample is adapted from [26].

**Figure 6 materials-15-08159-f006:**
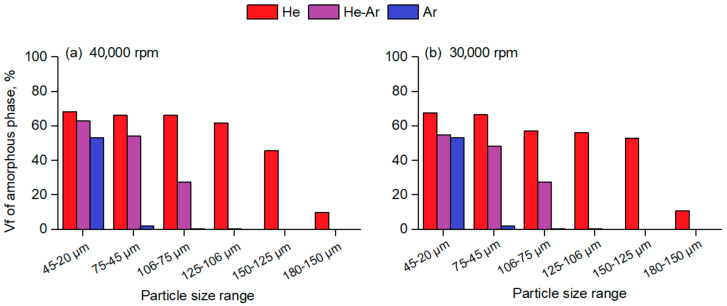
Amorphous volume fraction of the centrifugally atomized Al_86_Ni_8_Y_4.5_La_1.5_ powder as a function of the corresponding particle size range using different gas compositions: (**a**) particles atomized at a disk speed of 40,000 rpm; (**b**) particles atomized at a disk speed of 30,000 rpm.

**Figure 7 materials-15-08159-f007:**
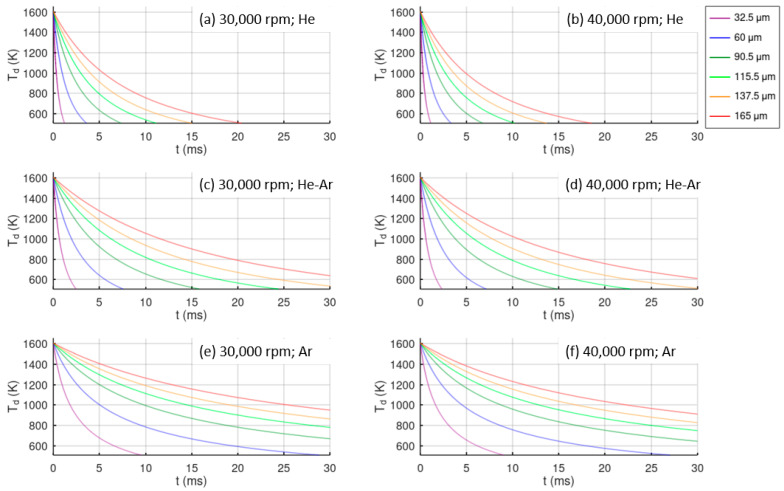
Temperature variation in centrifugally atomized droplets of different sizes in different atomization conditions of disk velocity and gas atmosphere.

**Figure 8 materials-15-08159-f008:**
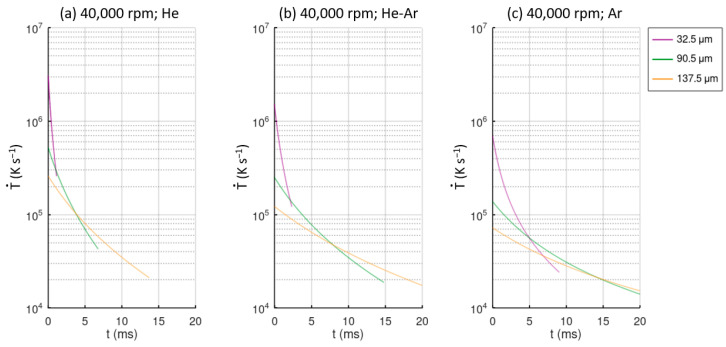
Cooling rate value over time for different particle sizes and different atomization conditions: (**a**) He atmosphere; (**b**) He-Ar atmosphere; (**c**) Ar atmosphere. All cases correspond to a disk velocity of 40,000 rpm.

**Figure 9 materials-15-08159-f009:**
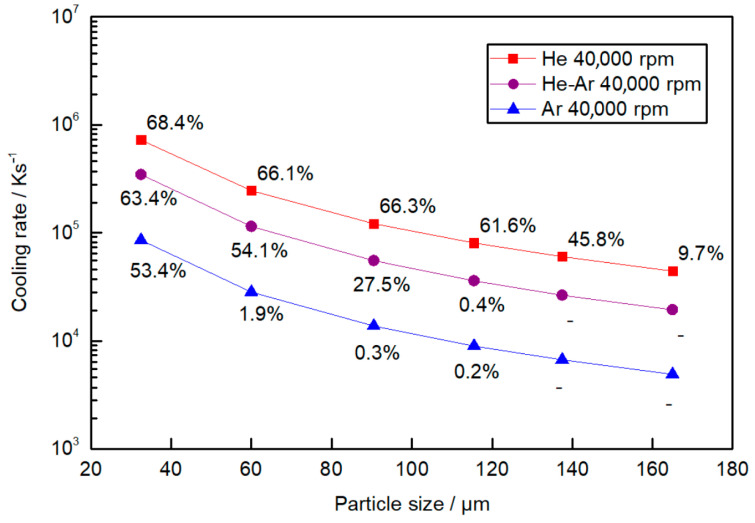
Cooling rate as a function of particle size atomized in a He, Ar, and He-Ar atmospheres at a disk speed of 40,000 rpm. The data points represent the amorphous fraction calculated from the DSC results for the corresponding mean particle size range.

**Table 1 materials-15-08159-t001:** Description of atomization runs and particle size distribution obtained.

Run #	Disc Velocity (rpm)	Gas Composition	D_10_ (µm)	D_50_ (µm)	D_90_ (µm)
1	30,000	He	63	111	165
2	30,000	He-Ar	77	136	231
3	30,000	Ar	78	127	193
4	40,000	He	55	96	156
5	40,000	He-Ar	59	102	155
6	40,000	Ar	51	90	150

## Data Availability

Data are contained within the article.

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
