# Peer review of "Centrifugal Atomization of Glass-Forming Alloy Al86Ni8Y4.5La1.5"

_materials, 2022, doi:10.3390/ma15228159_

Round 1

Reviewer 1 Report

1 The English level should be improved. For example, “Al-based metallic glasses have been proved very difficult to obtain…..” should be “Al-based metallic glasses have been proved to be very difficult to obtain….”

2 This is mainly due to thermal conductivity of He is greater than Ar (1.52·10-1 Wm-1k-1 value for He and 1.77·10-2 Wm-1k-1 for Ar, both at 273 K).” Why the authors provide the thermal conductivity at 273K rather than at room temperature? Here, the authors are suggested to give the initial temperatures of Ar and He for the centrifugal atomization.

3 What are the advantages of centrifugal atomization? The authors are suggested to highlight this point in Introduction.

4 The authors stated “For this work, the enthalpy of crystallization of the fully amorphous melt-spun ribbon sample was taken from Yang et al. and used as a reference”. Here, DSC curve of ribbon sample should be provided in Fig. 5, in order to convince the readers.

5 The journal names of all the references should be abbreviated, such as Powder Metallurgy, Journal of Sustainable Metallurgy, Philosophical Magazine, Rare Metal Materials and Engineering, Metallurgical and Materials Transactions B: Process Metallurgy and Materials Processing Science, Materials Science and Engineering A.

Reviewer 2 Report

In this work authors systematically studied the centrifugal atomization of Al86 Ni8Y4.5La1.5 metallic glass. The authors used XRD, SEM, and DSC experiments to characterize the powders. The authors also used a theoretical model to calculate the cooling rate of different powder sizes and gas atmospheres. The results were analyzed systematically and also discussed in depth. These results are exciting and novel enough to publish in the Materials, however, there are some minor issues needed to be addressed before the official acceptance.

 Minor revision Recommended

Comments to Author

·        In Figure 4, mark the phases of the XRD peaks.

·    In section 3.3 DSC analysis, lines 199-201, give the exact phase composition rather than a vague description.

·     Since the whole amorphous percentage calculation is based on the assumption that melt-spun ribbons are 100 % amorphous, it is recommended to give the DSC curves of the ribbons also in Fig 5 for a comparison.

 ·   Section 3.5 Thermal evaluation of Atomized droplets, the authors mentioned about semi-empirical equation based on Whitaker correlation, it is recommended to give the empirical equation in this section, along with all the parameters 

Round 2

Reviewer 1 Report

The authors have addressed the comments. I have no further comments and recommend accepting this manuscript for publication